# A Complement Method for Magnetic Data Based on TCN-SE Model

**DOI:** 10.3390/s22218277

**Published:** 2022-10-28

**Authors:** Wenqing Chen, Rui Zhang, Chenguang Shi, Ye Zhu, Xiaodong Lin

**Affiliations:** 1Innovation Academy for Microsatellites of Chinese Academy of Sciences, Shanghai 201203, China; 2University of Chinese Academy of Sciences, Beijing 100049, China

**Keywords:** magnetometer, actual magnetic data, stray magnetic, different working conditions, TCN-SE network

## Abstract

The magnetometer is a vital measurement component for attitude measurement of near-Earth satellites and autonomous magnetic navigation, and monitoring health is significant. However, due to the compact structure of the microsatellites, the stray magnetic changes caused by the complex working conditions of each system will inevitably interfere with the magnetometer measurement. In addition, due to the limited capacity of the satellite–ground measurement channels and the telemetry errors caused by the harsh space environment, the magnetic data collected by the ground station are partially missing. Therefore, reconstructing the telemetry data on the ground has become one of the key technologies for establishing a high-precision magnetometer twin model. In this paper, firstly, the stray magnetic interference is eliminated by correcting the installation matrix for different working conditions. Then, the autocorrelation characteristics of the residuals are analyzed, and the TCN-SE (temporal convolutional network-squeeze and excitation) network with long-term memory is designed to model and extrapolate the historical residual data. In addition, MAE (mean absolute error) is used to analyze the data without missing at the corresponding time in the forecast period and decreases to 74.63 nT. The above steps realize the accurate mapping from the simulation values to the actual values, thereby achieving the reconstruction of missing data and establishing a solid foundation for the judgment of the health state of the magnetometer.

## 1. Introduction

Geomagnetic navigation is a critical autonomous navigation method for near-Earth spacecraft, which has strong real-time, concealment, and interference suppression. Geomagnetic navigation uses the three-axis magnetometer installed on the spacecraft to measure the geomagnetic intensity in three directions that are not coplanar in space below 1000 km, and compares it with the nominal geomagnetic model to determine the orbit and attitude of the satellite [1]. Therefore, the health status of magnetometers is very important for satellite navigation and attitude determination. However, due to the limited capacity of the satellite–Earth measurement channels and the telemetry errors caused by the harsh space environment, the magnetic data are often missing, which makes it difficult to study the laws of magnetic data and establish accurate models using historical data. Thus, establishing an accurate complement model is a prerequisite for building a geomagnetic model.

The international geomagnetic reference model (IGRF) [2] and the world geomagnetic field model (WMM) are the commonly used geomagnetic models. They are based on the Gaussian spherical harmonic analysis theory, and building the expression between the geomagnetic vector elements and the geographical location. Then, the researchers comprehensively utilize the observation data from the marine magnetic survey, ground station magnetic survey, aeromagnetic survey, and satellite magnetic survey to solve the Gaussian coefficients of the model based on the standard least squares method. Although these models can comprehensively describe the distribution characteristics of the geomagnetic field, they ignore the local details such as the external field of the geomagnetic field and lithosphere. In addition, the geomagnetic measurement is affected by the stray current in different directions emitted by satellite components under various working conditions. Accordingly, the missing values cannot be directly supplemented by the IGRF and WMM models.

Marco et al. [3] predicted the subsequent missing points from the data before the missing points by the autoregressive moving average model, but this does not make full use of the subsequent data of the missing points and the overall utilization of the data is low. Zhu et al. [4] used the SVM algorithm to predict missing data, however, the SVM method cannot fill in a large number of missing data effectively. Considering the strong nonlinear fitting ability of the neural network, the authors of [5] established the BP neural network to predict the lost data and had effective results for filling the missing value. However, the BP network is not a time series model and it did not perform feature extraction and signal-to-noise ratio improvement, which is difficult to achieve high precision requirements. Li et al. [6] obtained the probability distribution of missing data by analyzing the statistical characteristics, and used the maximum likelihood to estimate the most suitable values.

Due to the periodic rotation of the satellite around the Earth, the data sampled by the sensors often have autocorrelation. Therefore, it is a better way to adopt the time series models with historical memory as the prediction model.

At present, the traditional autoregressive prediction technology has been mature, including autoregressive moving average (ARMA) [Baptista M ] and autoregressive integrated moving average (ARIMA) [7,8]. These statistical models are widely used in the industry. However, they can only be linearly expressed and the signal must be assumed to be stationary. The Kalman filter [9,10], least squares support vector machine (LS-SVM) [11,12], and other methods have been studied for a long time and the theories are mature. Therefore, they have been applied in many fields to solve time series prediction tasks. However, these methods can only deal with short-term autocorrelation tasks and are difficult to model long-term dependence of time series.

In recent years, with the rapid development of deep learning, the time series prediction method based on deep learning theory has been highly studied.

The authors of [13] used historical information of time series to predict future Turkish electricity load by a recurrent neural network (RNN). The authors of [14] adopted long short-term memory (LSTM) as a one-step prediction model for satellite telemetry data, and proposed an adaptive threshold algorithm to obtain the best threshold. It achieves telemetry data anomaly detection finally. The authors of [15] proposed a model combining a two-channel convolutional neural network (CNN) and LSTM. They used this model to predict short-term photovoltaic power and achieved better results. However, LSTM has the disadvantages of difficult model training, long training time, and unstable gradients in long-term dependence modeling. Temporal convolutional network [16] is a one-dimensional convolutional neural network. Compared with LSTM, TCN has more stable gradients, a flexible receptive field, and a shorter training time. Studies have shown that the network has achieved better results in audio synthesis [17], word language modeling [18], machine translation [19], and other fields.

In this paper, considering stray magnetic fields in different directions under various working conditions, the linear model is first used to correct the installation matrix. Moreover, since the residuals between the actual value and corrected simulation values in the installation coordinate system have autocorrelation, and to avoid the establishment of complex physical models and a large number of calculations, establishing a temporal model with historical memory ability is an effective and practical method that conforms to the residual characteristics.

## 2. Data Description and Calibrating Installation Matrix

### 2.1. Coordinate Systems

Before showing the data and calibrating method, we introduce several coordinate systems involved in this paper.

J2000 coordinate system: As shown in Figure 1, the origin is at the center of the Earth, the X axis points to the vernal equinox, the Y axis is in the equatorial plane and perpendicular to the X axis, and the Z axis points to the North Pole.Orbital coordinate system (VVLH): The origin of the orbital coordinate system is located at the satellite centroid, the Z axis points to the Earth center, the X axis is in the orbital plane and perpendicular to the Z axis, and the Y axis is determined by the right-hand rule.Body coordinate system: The origin of the body coordinate is located at the satellite’s center of mass, the X axis points to the satellite’s flight direction, the Z axis is in the satellite’s longitudinal symmetry plane, and the Y axis is perpendicular to the satellite’s longitudinal symmetry plane.Installation coordinate system: The installation position of the magnetometer varies from satellite to satellite. The three-axis installation coordinate system of the magnetometer is obtained by multiplying the body coordinate system by the installation matrix. The coordinate system is shown in Figure 2.

### 2.2. Geomagnetic Field Measurement

The geomagnetic field measurement can be expressed as,
(1)$$\begin{array}{cc} M_{A}\; & =\left(R_{AB}*R_{BO}*R_{OI}\right)*M_{IGRF}+\tilde{\xi }\\ \; & =\left(\hat{R_{AB}}+\tilde{R_{AB}}\right)*\left(\hat{R_{BO}}+\tilde{R_{BO}}\right)*\left(\hat{R_{OI}}+\tilde{R_{OI}}\right)*\left(M_{IGRF}+{\tilde{M}}_{IGRF}\right)+\tilde{\xi } \end{array}$$
where $M_{A}$ and $M_{IGRF}$ are the magnetic vectors under the installation coordinate system (the fluxgate magnetometer measurement coordinate system) and J2000 inertial coordinate system (obtained by the IGRF model, which is established in Satellite Tool Kit), respectively. ${\tilde{M}}_{IGRF}$ is the influence of the lithosphere and external field, which is not considered in the IGRF model. $R_{OI}$, $R_{BO}$, and $R_{AB}$ are the transformation matrices of the inertial coordinate system to the orbital coordinate system, orbital coordinate system to body coordinate system, and body coordinate system to the installation coordinate system, respectively. $\hat{R_{AB}}$, $\hat{R_{BO}}$, and $\hat{R_{OI}}$ are the main parts of the three which are evaluated at the ground station and $\tilde{R_{AB}}$, $\tilde{R_{BO}}$, and $\tilde{R_{OI}}$ are the errors. $\tilde{\xi }$ is the error term. Since the orbit error is less than 10 m and the attitude determination precision of the star tracker is on the order of angular seconds [20], the $\tilde{R_{BO}}$ and $\tilde{R_{OI}}$ can be ignored. The above formula can be simplified as,
(2)$$M_{A}=\left(\hat{R_{AB}}*\hat{R_{BO}}*\hat{R_{OI}}+\tilde{R_{AB}}*\hat{R_{BO}}*\hat{R_{OI}}\right)*\left(M_{IGRF}+{\tilde{M}}_{IGRF}\right)+\tilde{\xi }$$
where $\hat{R_{AB}}$, $\hat{R_{BO}}$, and $\hat{R_{OI}}$ are not affected by working conditions and are the determined matrices. Excluding the considered part, the residual error $M_{RS}$ (as shown in Figure 3) is,
(3)$$\begin{array}{cc} M_{RS} & =M_{A}-\hat{R_{AB}}*\hat{R_{BO}}*\hat{R_{OI}}*M_{IGRF}\\ & =\hat{R_{AB}}*\hat{R_{BO}}*\hat{R_{OI}}*{\tilde{M}}_{IGRF}+\tilde{R_{AB}}*\hat{R_{BO}}*\hat{R_{OI}}*M_{IGRF}\\ & +\tilde{R_{AB}}*\hat{R_{BO}}*\hat{R_{OI}}*{\tilde{M}}_{IGRF}+\tilde{\xi } \end{array}$$

The sources of residual error are as follows. The multiplicative or additive combination of these influences constitutes the residual error.

Satellite body components: The current of solar panel, battery and magnetic torque, and hard and soft magnetic materials on the satellite generate magnetic fields [21]. These disturbances change with attitude, showing a time-related regularity. This part is the source of stray magnetism.External magnetic interference (not included in IGRF model): (1) The local anomaly of the lithosphere: it mainly comes from the short wavelength magnetic field caused by the geological characteristics of the Earth’s surface (mountains, oceans, minerals, etc.) [22]. (2) External field: the space current system above the Earth’s surface, which is mainly distributed in the ionosphere, that is, the upper atmosphere ionized by solar high-energy radiation and cosmic ray excitation [23]. The impact of this item can reach dozens of nT [24]. These two terms show periodic changes with orbital periods.IGRF fitting error. This part is minimal. Ref. [25] shows that the accuracy of the spherical harmonic coefficient of the eighth-order IGRF-11 is 0.1 nT.Effects of temperature change: (1) Uneven expansion of the satellite and the changes in the installation coordinate system, which has a slight impact. (2) Thermal drift, which is relatively large. Ref. [26] shows that the maximum drift of the magnetometer reaches 200 nT between 0 and 40 °C. This effect of temperature changes regularly with the satellite in the Earth’s shadow area or sunshine area.

### 2.3. Data Description

The data processed in this paper are all three-axis vector moduli of the geomagnetic field. In this paper, a total of 10 days of on-orbit magnetic data are collected and 5722 discrete points are obtained by a sampling frequency of 128 s. Figure 4 shows the actual sampled data, and the enlarged subfigure shows part of the modulus of the actual data and the simulation data in the installation coordinate system (before calibrating). The overall varying ranges are [15,492.92 nT, 42,431.39 nT] and [17,769.52 nT, 44,356.67 nT], respectively.

Because of the complex space environment, remote sensing errors may happen, which results in abnormal values marked in red circles, as shown in Figure 4. The steps for handling such outliers are as follows,

Use the method in Algorithm 1 to detect abnormal sampling points:The outliers detected in step 1 are deleted, and then supplemented by our imputation method proposed in Figure 5.

  **Algorithm 1**: 3σ outlier detection method

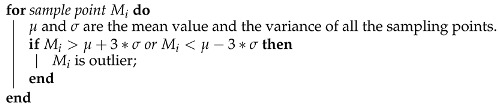



Moreover, due to the missing points caused by communication blocking, 505 points are missed, and the maximum missing interval is 24 points, which will provide a basis for the following model parameter setting.

Figure 3 shows partial residual errors described in Formula (3). The overall range is [−537.97 nT, 1644.46 nT]. Ref. [27] used the extended Kalman filtering method to simulate the real magnetic measurement data of ERBS and GRO satellites, and the orbit determination error was 10–40 km under the condition of measurement error of 400 nT. Based on engineering experience, the error range obviously cannot meet the engineering requirements.

When the satellite directs the Earth and the Sun, respectively, the satellite body temperature is different, resulting in different degrees of thermal deformation and different effects on $\tilde{R_{AB}}$ and $\tilde{\xi }$. At the same time, the difference in satellite attitude makes the stray magnetism in different directions have different effects on $\tilde{\xi }$ under the installation system. Therefore, we first deal with the additive and multiplicative errors caused by working conditions in Formula (3).

### 2.4. Calibration of Installation Matrix

The corresponding times of directing the Sun and the Earth are obtained from the satellite orbit model established in STK. As for the time in the conversion period, it only accounts for 0.17% in one period, and the corrected simulation values in the installation coordinate system of the conversion period by calibrating directing Sun installation matrix do not generate large errors. So we will not calibrate separately. In this paper, three multivariate linear regression models are used to correct the installation matrix corresponding to the X-Y-Z three-axis vectors, and the overall format is (matrix format),
(4)$$\left(\begin{array}{c} \hat{M_{A\_X}}\\ \hat{M_{A\_Y}}\\ \hat{M_{A\_Z}} \end{array}\right)={\hat{R_{\mathrm{AB}}}}^{\prime }*\hat{R_{\mathrm{AB}}}*\hat{R_{\mathrm{BO}}}*\hat{R_{\mathrm{OI}}}*\left(\begin{array}{c} M_{\mathrm{IGRF}\_X}\\ M_{\mathrm{IGRF}\_Y}\\ M_{\mathrm{IGRF}\_Z} \end{array}\right)+\left(\begin{array}{c} \xi_{X}\\ \xi_{Y}\\ \xi_{Z} \end{array}\right)$$
where [$\hat{M_{A\_X}},\hat{M_{A\_Y}},\hat{M_{A\_Z}}$]^*T*^ are the actual value of the three-axis magnetic vector, [$M_{\mathrm{IGRF}\_X}$, $M_{\mathrm{IGRF}\_Y}$, $M_{\mathrm{IGRF}\_Z}$] are the three-axis magnetic vector under the inertial system, and [$\xi_{X}$, $\xi_{Y}$, $\xi_{Z}$] are the error vectors. All the above vectors are one-dimensional data of 1×n, and *n* is the number of actual values. The ${\hat{R_{\mathrm{AB}}}}^{\prime }$ is the calibrated installation matrix (3×3) from the body system to the installation system, and [$\hat{R_{\mathrm{AB}}}$, $\hat{R_{\mathrm{BO}}}$, $\hat{R_{\mathrm{OI}}}$] are equal to the ones in use and are all sizes of (3×3). The three linear regression models follow the assumptions:The three-axis magnetic vectors [$M_{\mathrm{IGRF}\_X}$, $M_{\mathrm{IGRF}\_Y}$, $M_{\mathrm{IGRF}\_Z}$] are not related to each other;The [$\xi_{X}$, $\xi_{Y}$, $\xi_{Z}$] are independent and identically distributed;The three-axis vectors [$M_{\mathrm{IGRF}\_X}$, $M_{\mathrm{IGRF}\_Y}$, $M_{\mathrm{IGRF}\_Z}$] are not related to the random term [$\xi_{X}$, $\xi_{Y}$, $\xi_{Z}$], that is, $Cov(M_{\mathrm{IGRF}\_\mathrm{ij}},\xi_{i})=0$, i∈[X,Y,Z], j=1,2,…,n.

Since there is no analytical solution when [$M_{\mathrm{IGRF}\_X}$, $M_{\mathrm{IGRF}\_Y}$, $M_{\mathrm{IGRF}\_Z}$] × [$M_{\mathrm{IGRF}\_X}$, $M_{\mathrm{IGRF}\_Y}$, $M_{\mathrm{IGRF}\_Z}$]^*T*^ is not an invertible matrix, it takes a lot of time when the number of data increases, especially when calculating the inverse matrix. So we use SGD (stochastic gradient descent). The SGD algorithm is an improved algorithm based on gradient descent. It randomly selects a sample to iterate once, rather than for all samples. Therefore, Algorithm 2 significantly reduces the calculation magnitude.    
**Algorithm 2**: The SGD algorithm (take the X axis as an example)
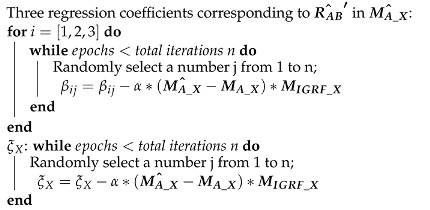


The data sampled is divided into the training set and the test set by the ratio of 8:2. The training data are used to fit the installation matrix. The test set is used to test the calibration effect.

After correcting the installation matrix by the above steps, the residual range between the actual value and the simulation value translated to the installation coordinate system is in [−540.55 nT, 511.98 nT] in the test set. It can be seen from Figure 6 that this step significantly eliminates the influence of different working conditions and greatly reduces the error range.

However, the linear regression only assumes the linear relationship between the dependent variable and the independent variable, and cannot directly deal with the multiplicative errors described in Section 2.2 (these errors also cannot be expressed by mathematical formulas with clear physical meaning) and high-order small quantities. In addition, the influence of local lithospheric anomaly and external field are not included in the IGRF model, and this step cannot deal with this impact. Therefore, we consider establishing a nonlinear model to further reduce the error by learning the rules of these residuals by using the characteristics of the residual sequence.

## 3. Processing of Residual Errors

### 3.1. Autocorrelation of Residual Sequence

The residual sequences of different working conditions after calibration are obtained from Section 2.4, and then spliced in chronological order. Then the autocorrelation of the observation sequence is calculated by using the autocorrelation function. The autocorrelation function is defined as,
(5)$$\rho_{k}=\frac{\sum_{t=1}^{n-k}\left(RS_{t}-\bar{RS}\right)\left(RS_{t-k}-\bar{RS}\right))}{\sum_{t=1}^{n}{\left(RS_{t}-\bar{RS}\right)}^{2}}$$
where $RS_{t}$ is the *t*-th value of residual sequence RS, $\bar{RS}$ is the mean value of sequence RS, *k* is the lag order, and *n* is the sequence length.

Figure 7 shows the calculation result. The dashed line represents the 99% confidence intervals. When the autocorrelation value corresponds to a lag order located outside the dashed line, it indicates that the lag order has autocorrelation in statistical significance. It can be seen from the figure that the residual sequence remains the maximum 135-order autocorrelation (since the subsequent orders do not exceed the confidence intervals, only the first 1000 orders are shown).

Therefore, we seek a time series model with long-term historical memory ability to decrease sequence autocorrelation to reduce residuals.

### 3.2. Time Series Model

LSTM is a commonly used model with long and short-term memory ability. Zhang et al. [28] adopted long short-term memory to reflect long-term memories of battery degradation tendency. Lu et al. [29] predicted electricity price based on LSTM. However, there is a risk of gradient vanishing and gradient explosion, and this is time-consuming due to the property of the back-propagation direction of time. A detailed discussion is shown in Appendix A. Hence, we use the TCN to avoid the disadvantages of LSTM and meet the needs better:

#### 3.2.1. TCN Network

Dilated Causal ConvolutionAs shown in Figure 8, TCN has the characteristics of both causal convolution and dilated convolution [30]. Formally, for one-dimensional input sequence RS and a filter f:0,…,k−1→R, the convolution result is,
(6)$$F\left(t\right)=\left(RS{*}_{d}f\right)\left(t\right)=\sum_{i=0}^{k-1}f\left(i\right)\cdot RS_{t-d*i}$$
where *k* is the convolution filter size, *d* represents the sampling interval (expansion coefficient), and t−d∗i accounts for the direction of the past. When d=1, it indicates that each point is collected, and the higher the level, the larger the *d*.Causal convolution ensures that the output result at each time is determined by the current and previous time. The advantage of dilated convolution is that it increases the receptive fields without pooling loss information, so that each convolution output contains a large range of information. At the same time, the property of interval sampling avoids the problem of too many convolution layers and reduces the risk of gradient disappearance. Thus, it allows for both very deep networks and a very long effective history.Residual connectionResidual connection transmits information directly between layers without convolution, which is a very effective way to train deep networks [31]. Figure 9 shows that in a residual block, the signal is transmitted by a weight layer composed of a series of operations such as causal expansion convolution and normalization, and also is directly mapped to the sum of the output of the weight layer,
(7)o(t)=RS(t)+F(t)
where *o* is the output of the residual block. If F(t)=0, the block turns into an identity map, and thus degenerates into a shallow network. This structure has the advantages of both the depth of the deep network and the shallow network. At the same time, even if the gradient of the weight layer is 0, the gradient of the entire residual network can be guaranteed not to be 0 due to the existence of the direct mapping mechanism. This block reduces the risk of vanishing gradients further when the model depth increases.The advantages of the TCN can be analyzed as follows:In backpropagation, gradients do not accumulate along the time dimension, but along the network depth direction, and the number of network layers is often much smaller than the length of the input sequence. Furthermore, with the help of residual connection, the risk of gradient disappearance is further reduced, thus avoiding the gradient vanishing of RNN and LSTM networks.The TCN does not have to wait until the previous time is completed before calculating the result of the current time. Instead, all times are convoluted at the same time, which realizes parallelization and greatly reduces the running time.The dilated casual convolution mechanism of the TCN determines that it can use long-term historical information to predict by increasing the receptive field without dramatically increasing the model depth, which proves that it has the ability to remember long-term historical information.In addition, in the copy memory task experiment [16], the TCN has no decline in memory ability in the sequence length from 0 to 250, and the 135 orders autocorrelation of our residual sequence is just in this range.Formula of the TCN receptive field(8)$$R_{field}=1+2*\left(k-1\right)*N_{stack}*\sum_{i}d_{i}$$
where $N_{stack}$ represents the number of residual blocks in the TCN, and the sum of $d_{i}$ indicates how many causal convolution layers are stacked in each residual block. For example, d=[1,2,3,4] indicates that four causal convolution layers are stacked, and the expansion factors of each convolution layer are 1, 2, 3, and 4, respectively.We determine the parameters of the TCN according to the above formula and Figure 8, the sequence has the maximum 135-order autocorrelation, so the input sequence length is set to 150; since the first five orders have significant autocorrelation (the autocorrelation coefficients are all greater than 0.5), the convolution kernel size is set to 6; the length of the input sequence is not too large, so the number of residual blocks is set to 2. This way the receptive field is larger than the length of the input sequence. According to Formula (6), the depth is set to 4, d=[1,2,4,8]. The maximum missing points interval is 24, so the output length of the whole model is set to 30.

#### 3.2.2. SE Attention Mechanism Module

SENet is a network model in computer vision, namely the squeeze-and-excitation network, which contains an attention module. It can learn the correlation between the output channels of the TCN module, assigns a weight to each feature channel, and pays more attention to those channels with key features, so as to improve the feature extraction ability of our method and the performance of the model. The SENet has a small amount of calculation and is very suitable for embedding into various networks. It mainly includes the steps of squeeze and excitation. See Appendix B for a detailed description.

## 4. The Overall Framework

The overall framework is shown in Figure 5. Firstly, we calibrate the installation matrix by the simulation data and actual data of different working conditions. Then we concatenate the residual errors between actual values and corrected simulation values in the installation coordinate system in chronological order for training the TCN-SE model. For the breakpoint t and the following *p* consecutive breakpoints, we correct the 1+p simulation data under the installation coordinate system by the corrected installation matrix first, then add the corresponding residuals predicted by the trained model, and the total values are the complements for the breakpoint *t* and the following *p* breakpoints.

## 5. Experiment and Result Analysis

### 5.1. Continuous Sequence

#### 5.1.1. Data Acquisition

For the residual sequence obtained in Section 2.4, select the point that meets the following requirement: 179 points after this point are uninterrupted. Among them, 150 points are used as model input, and 30 points are used as model output. Including this point, 180 sampling points are added to the continuous sequences. All such sequences are divided into training sets and testing sets at a ratio of 8:2. For the sampling point *n*, such a continuous sequence form is,
(9)$$continuous\_sequence\_n=[x_{n},x_{n+1},\ldots ,x_{n+149},x_{n+150},\ldots x_{n+179}]$$

#### 5.1.2. Model Evaluation Metrics

To evaluate the performance of the model, we use the MAE and the error range. The MAE for continuous sequence *n* (abbreviated to CSn) is defined as,
(10)$$MAE=\frac{\sum_{n=1}^{N}\sum_{t=150}^{179}|CSn_{t}-\hat{CSn_{t}}|}{30*N}$$
where $CSn_{t}$ and $\hat{CSn_{t}}$ represent the predicted value and the real value of the continuous sequence n at time *t*, respectively. *N* is the number of continuous sequences. The smaller the MAE, the better the performance of the model.

The error range is defined as,
$$\begin{array}{cc} \; & error\;range=(Minimum\left(CSn_{t}-\hat{CSn_{t}}\right),Maximum\left(CSn_{t}-\hat{CSn_{t}}\right))\\ \; & t\in [150,151,\ldots 179],n\in [1,2,\ldots N] \end{array}$$

#### 5.1.3. Deep Learning Experiment

All the experiments were based on Windows operating system. Tensorflow (version 2.4.0) was selected as the deep learning framework. The GPU is NVIDIA RTX6000.

According to Formula (8), in order to make the receptive field cover the autocorrelation orders, we set the TCN-SE network parameters and training parameters as in Table 1:

#### 5.1.4. Comparison of Different Models

The MAE and error range are used as evaluation indicators. The comparison of the prediction effects of LSTM, TCN, and TCN-SE models on the testing set is shown in Table 2. Compared with LSTM and TCN, the TCN-SE has significantly reduced MAE, error range, and time consumption.

#### 5.1.5. Prediction on Test Set

To compare with actual values, the predictions in the testing set are added by corrected simulation values in the installation coordinate system by the correction installation matrix from Section 2.4 of the corresponding time. Figure 10 and Figure 11 show comparison of prediction effects on different days. The selected subfigures show residual errors of three models for the corresponding time. The subfigures show residual errors of three models in the selected time periods, which are always difficult to predict accurately. It can be seen that our method is accurate at almost every sampling point in the two figures, and has stronger long-term memory ability and more reasonable distribution mechanism.

### 5.2. Sequence with Breakpoints

#### 5.2.1. Data Acquisition

For the residual sequence obtained in Section 2.4, we select the breakpoint that meets the following requirement: the number of consecutive breakpoints after the breakpoint is less than 30. The first 150 sampling points before the breakpoint and the following 30 sampling points including the breakpoint are added to the sequences. For the breakpoint *m*, it is assumed that *p* points are continuously missing after the breakpoint and expressed by ∗. The form of the sequence with breakpoint *m* (abbreviated to SWBm) is,
(11)$$SWBm\;=\;[x_{m-150},x_{m-149},\ldots ,x_{m-1},*,*,\ldots ,x_{m+p},\ldots x_{m+29}]$$

#### 5.2.2. Model Evaluation Metrics

The MAE for sequences with breakpoints is defined as,
$$MAE=\frac{\sum_{m=1}^{M}\sum_{t=p_{m}+150}^{179}|SWBm_{t}-\hat{SWBm_{t}}|}{\sum_{m=1}^{n}30-p_{m}}$$
where $SWBm_{t}$ and $\hat{SWBm_{t}}$ are, respectively, expressed as the real values and the model predicted values of breakpoint sequence *m* at time *t*. $p_{m}$ is the number of breakpoints in sequence *m*. *M* is the number of sequences with breakpoints. The smaller the MAE, the better the performance of the model.

The error range is defined as,
$$\begin{array}{cc} \; & error\;range=(Minimum\left(SWBm_{t}-\hat{SWBm_{t}}\right),Maximum\left(SWBm_{t}-\hat{SWBm_{t}}\right))\\ \; & t\in \left[p_{m}+150,p_{m}+151,\ldots 179\right],m\in \left[1,2,\ldots M\right] \end{array}$$

#### 5.2.3. Comparison of Different Models

Since the missing points cannot be supplemented, we use the predictions after the breakpoints in the forecast period, which are at the time without missing values to demonstrate the extrapolation and the complementation ability of the method. In addition, there is a little difference from before. The predicted residuals at the time of missing points are added to the actual residual sequence as historical residual values, to avoid the situation that the number of continuous historical data before some breakpoints is less than the model input length. At the same time, we realize rolling complement and prediction. The MAE and error range in Section 5.2.2 are used as evaluation indicators. The comparison of the prediction effects of LSTM, TCN, and TCN-SE models which are trained on continuous sequences on the test set is shown in Table 3. We found a similar conclusion to that in Section 5.1.4.

#### 5.2.4. Prediction Effect

Similar to Figure 10, the orange curve in Figure 12 and Figure 13 is the actual values. The blue breakpoints consist of the sum of 30 (model output length) predictions after breakpoints and corrected simulation values in the installation coordinate system of corresponding time. The subfigures show the residual errors in the corresponding periods.

## 6. Conclusions

In this paper, a two-step model is proposed, which first corrects the installation matrix under different working conditions and uses the deep learning method to fit the residual errors. Considering the different effects of the external field, lithosphere, and stray magnetic field in different directions, the installation matrix is corrected separately in two working conditions. Because the residual sequence contains autocorrelation and nonlinear information, the TCN network with dilated causal convolution mechanism is used for fitting, and the SE model with attention mechanism is used to improve the accuracy. Experiments show that the TCN-SE used in this paper has an MSE of 74.63 nT and an error range of [−215.18 nT, 211.63 nT] when predicting the subsequent data of missing values. The extrapolation ability of our method is proved, and it can be used to complement the missing values of on-orbit magnetic data.

However, the installation matrix is not only affected by two working conditions of directing the Sun and the Earth, but may also be related to the disturbance caused by different tasks (working conditions). In order to get more detailed classifications of working conditions and further reduce the linear error of the installation matrix, we consider clustering the residual sequence between the actual values and the inertial coordinate system values after being multiplied by the installation matrix while directing the Sun and the Earth time, respectively.

The universal magnetic map updates every five years, its magnetic isoplane moves 10–15 km every year, and the accumulated error will exceed 100 km in 10 years [32]. We consider that the special magnetic model established in this paper can also replace the magnetic map-matching method. It updates regularly according to the updated telemetry data, and can be used for magnetic navigation and satellite attitude determination after the extrapolated data are interpolated to meet the actual frequency requirements.

## Figures and Tables

**Figure 1 sensors-22-08277-f001:**
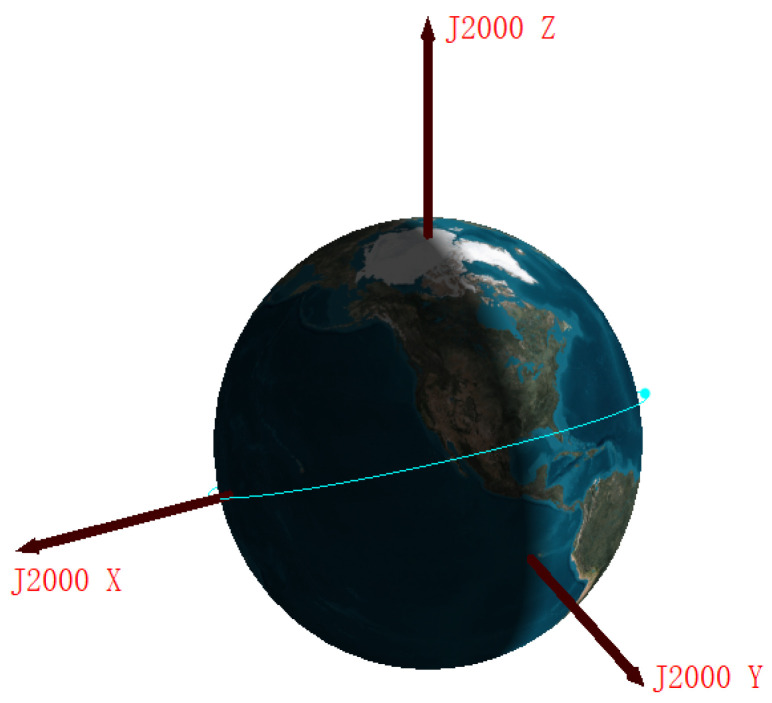
J2000 coordinate system. The cyan line is the satellite orbit.

**Figure 2 sensors-22-08277-f002:**
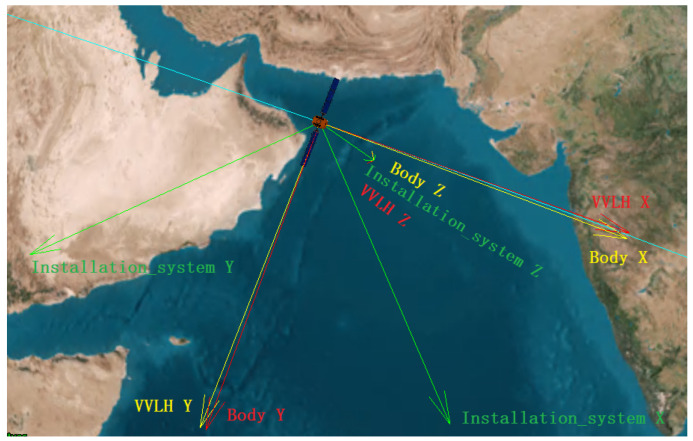
Orbital coordinate system, body coordinate system, and magnetometer installation coordinate system.

**Figure 3 sensors-22-08277-f003:**
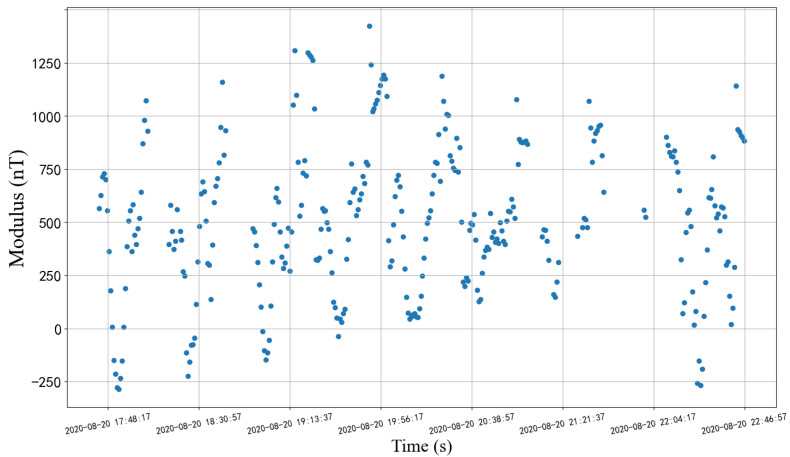
The residuals between the actual data and the simulation data in the installation coordinate system correspond to the subfigure in Figure 4.

**Figure 4 sensors-22-08277-f004:**
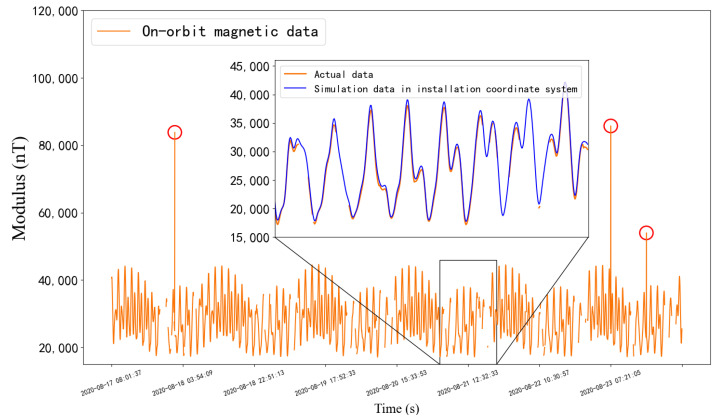
The father graph shows the on-orbit magnetic data. We select one segment for showing details of the actual data and the simulation data in the installation coordinate system, which are labeled in orange and blue curves, respectively.

**Figure 5 sensors-22-08277-f005:**
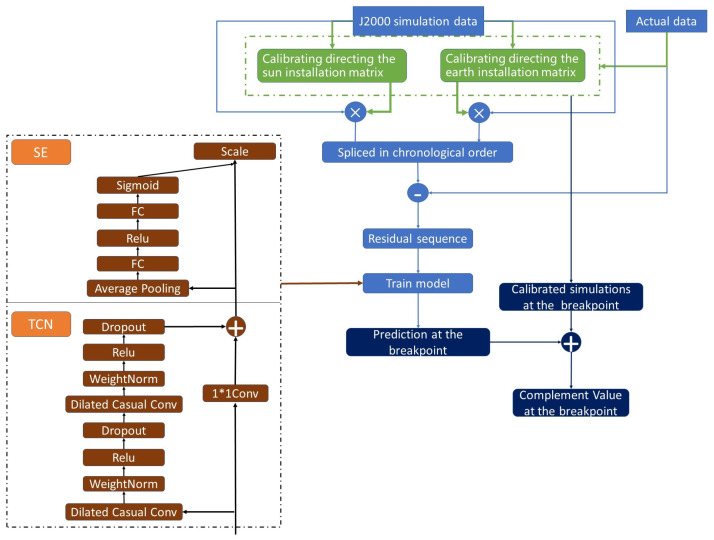
Complement algorithm flow chart.

**Figure 6 sensors-22-08277-f006:**
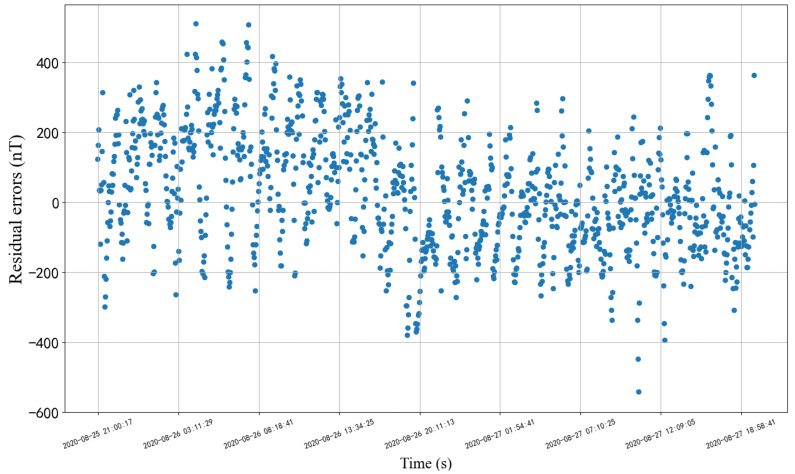
Residuals after calibrating installation matrix.

**Figure 7 sensors-22-08277-f007:**
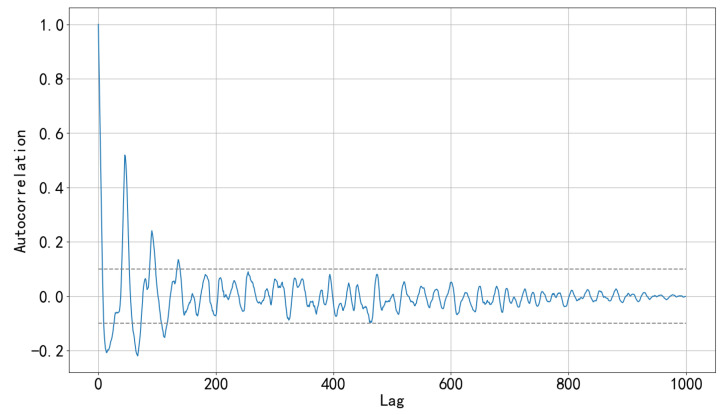
Sequence autocorrelation.

**Figure 8 sensors-22-08277-f008:**
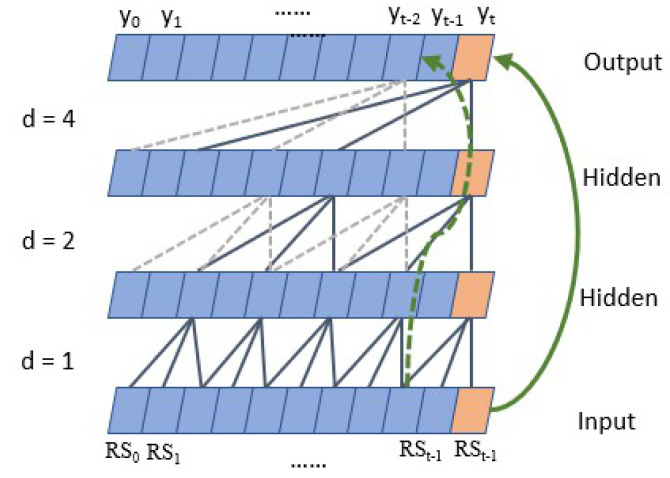
Dilated causal convolution. The green dashed and solid lines are residual connections. The blue and grey lines indicate dilated casual convolution method.

**Figure 9 sensors-22-08277-f009:**
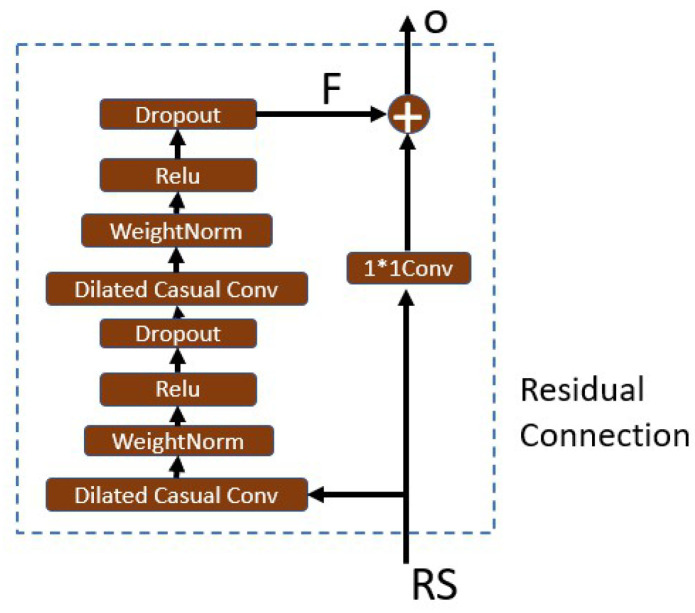
TCN residual block.

**Figure 10 sensors-22-08277-f010:**
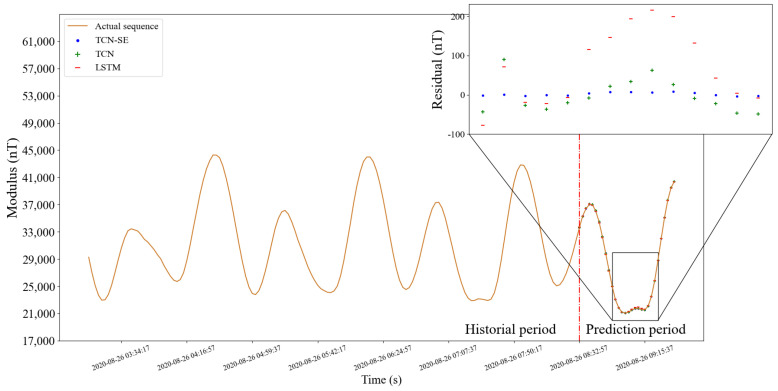
Prediction effect of three models. The orange curve is the actual sequence. The blue point, green plus symbol, and red horizontal line represent the predictions of the TCN-SE, TCN, and LSTM, respectively. We enlarged part of the forecast period to show the details of the residual errors of three models. Before and after the red dotted line are the historical and prediction periods, respectively.

**Figure 11 sensors-22-08277-f011:**
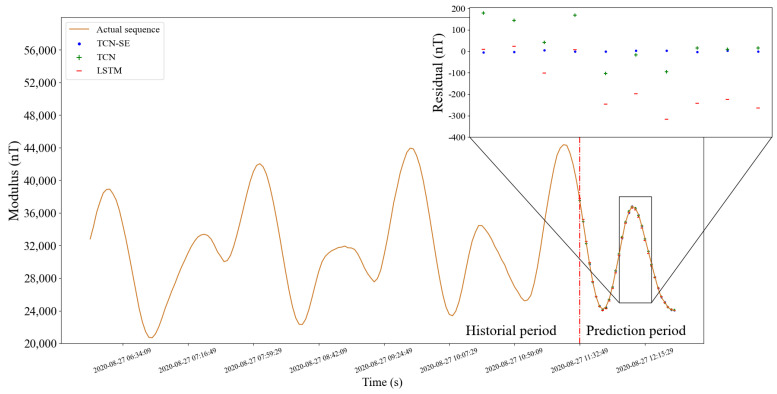
Prediction effect of the continuous sequence on different days from the above figure.

**Figure 12 sensors-22-08277-f012:**
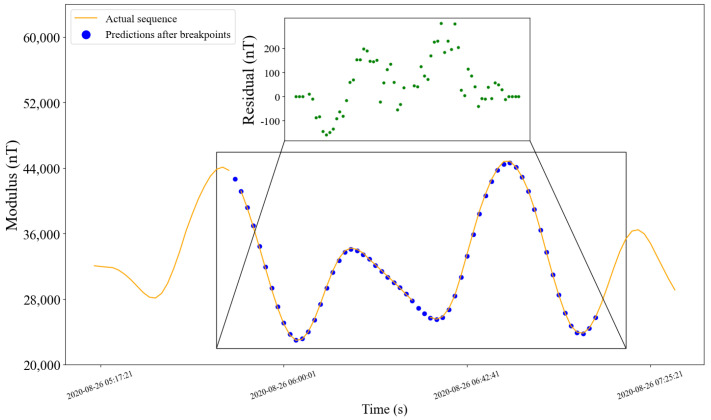
Prediction effect of the sequence with breakpoints. The orange curve is the actual data. Dark blue points are predictions after breakpoints to highlight locations. The subfigure shows the residual errors of the two curves in the selected period.

**Figure 13 sensors-22-08277-f013:**
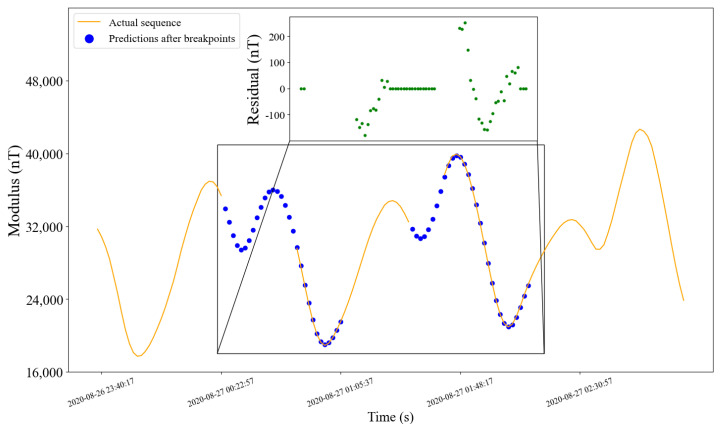
Prediction effect of the sequence with breakpoints on different days from the above figure.

**Table 1 sensors-22-08277-t001:** Model parameter settings.

Parameter	Settings
Number of residual blocks	2
Convolution kernel size	6
Dilations	[1, 2, 4, 8]
Model input length	150
Model output length	30
Number of iterations	100
Batch size	16
Loss function	MSE
Optimizer	Adam

**Table 2 sensors-22-08277-t002:** Model effect comparison.

Network	MAE (nT)	Error Range (nT)	Time Cost (s)
LSTM	112.39	(−446.24, 411.46)	545
TCN	67.27	(−225.98, 213.26)	176
TCN-SE	55.71	(−175.18, 168.28)	251

**Table 3 sensors-22-08277-t003:** Model comparison.

Network	MAE (nT)	Error Range (nT)
LSTM	144.15	(−483.42, 442.98)
TCN	92.19	(−254.67, 259.26)
TCN-SE	74.63	(−215.18, 211.63)

## Data Availability

Not applicable.

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
