# Peer review of "A Complement Method for Magnetic Data Based on TCN-SE Model"

_sensors, 2022, doi:10.3390/s22218277_

Round 1

Reviewer 1 Report

The article is on a very good level.

The paragraph between lines No89 and No90 is not labelled with numbers. This can be seen in some other paragraphs as well.

Regarding figure No1, I recommend enlarging the description of the X axis.
I also recommend enlarging the heading of the Y axis and placing a space between the description and the unit of this axis, in this very figure.
The Y axis of its subfigure should be enlarged as well.

There is a typing error in the word "segment" located in the description of figure No1.

There is a typing error in the word "actual" located in the description of figure No2.

I recommend enlarging the axis X and Y descriptions in figure No2, as well as placing a space between the heading and the unit.

There are typing errors in the words "residual" and "decrypted" found on line No106.

I also recommend adjusting the placement of the text "Residual Connection" in figure No6, so it does not overlap with the present line.

It would be beneficial to describe chapters No4. and No5.1.5 a bit more in-depth.

Do not let the texts "Residual", "Historial", and "Prediction Period" touch and overlap with borderlines in figures No8 and No9.

Enlarging the description of the X axis and not letting the text "Residual" touch/overlap with borderlines would be desirable in figures No10 and No11.

Reviewer 2 Report

The manuscript submitted by Chen et al. discusses the data correction and prediction algorithms based on the temporal convolutional network (TCN), to complement the missing satellite magnetometry data.  Although I do see general improvement over the previous methods (e.g., Figs. 8 and 9), I had hard time understanding the value of this study since the authors do not provide clear criteria for good/bad values.  I am not an expert in satellite navigation, and I believe that is the case for the majority of the Sensors journal readership.  In addition, the manuscript has multiple flaws in terms of punctuations, citations, and the unit of physical values, well beyond typical typos, and is not considered a carefully prepared manuscript.  Therefore, I strongly recommend addressing the following issues before considering this manuscript for publication.

-          Please provide the requirement for this type of measurement; what value is acceptable/non-acceptable?  Without that, it’s impossible to judge if the proposed method is satisfactory.  The authors wrote in line 107 “obviously cannot meet the engineering requirements” without providing those requirements.  What I can say based on given information is just it is “better” than the previous methods.

-          I request the authors to provide a figure that illustrates the three coordinate systems discussed in the manuscript, and the sources of measurement errors (which is called “stray magnetic field”).

-          If I understood correctly, the effect of temperature is considered/corrected only on the geometrical deformation; how about the sensor reading itself under significant temperature variations?  Actually, what kind of magnetic sensor is used?

-          According to, for example, Fig. 10, the proposed method works very well, to the point one may not need an actual measurement.  Obviously that is not the case and it requires model calibration from time to time, but the duration between each measurement (or amount of permissible missing data) can be much longer (or larger) than the current limit.  Please discuss those limits.

-          Revise the caption of figures; it should be something like “Prediction effect of the sequence with breakpoints.” instead of “The figure shows prediction effect of the sequence with breakpoints.”  This is the case throughout the manuscript.

-          Put a space between numbers and units.

-          Put a space between words and following parenthesis (e.g., “SGD (Stochastic)” instead of “SGD(Stochastic)”).

-          The unit of the magnetic flux density is T or nT, not nt.  And do not use Italic for unit.

-          TCN is first defined in line 76, then re-defined in line 164.

-          Separately, in the abstract, define TCN, SE, and MAE.

-          Line 81; “the influence of stray magnet” shall be “stray magnetic field”.

-          Line 85; “number of calculations. Establishing…” shall be “number of calculations, establishing”

-          Line numbers are missing between 89 and 90.  There are many other locations with missing line numbers.

-          Just before Section 2.2, “M_RS (as shown in Figure 5)” is referencing an incorrect figure, I believe.

-          Line 111; “permanent magnetic material” shall be “hard magnetic material”, which is used to create a permanent magnet.

-          Section 2.3 title; need “of” between “calibration” and “installation”.

-          Section 2.3, first paragraph; “So we will not calibrated separately” revise English to use correct grammar.

-          Equation (4) starts with unnecessary curly bracket.

-          Line 127; “invertible matrix. It takes” shall be “invertible matrix, it takes”.

-          Figure 3; plot around zero (i.e., vertical axis shall be [-200, 0, 200…] instead of [-150, 50, 250…]), so that the bias becomes obvious. 

-          Line 199; missing reference number.

-          Lines 315 and 317; put reference numbers in square brackets (and list the details in the bibliography section, of course) rather than the name of the first author.
